# MOBIEDIT: RESOURCE-EFFICIENT KNOWLEDGE EDITING FOR PERSONALIZED ON-DEVICE LLMS

**Zhenyan Lu[1,2], Daliang Xu[1], Dongqi Cai[1,4], Zexi Li[5,6], Wei Liu[7], Fangming Liu*[2],
Shangguang Wang[1], Mengwei Xu*[1,3]**
[1]Beijing University of Posts and Telecommunications, [2]Pengcheng Laboratory,
[3]Beiyou Shenzhen Institute, [4]Nanjing University, [5]The Chinese University of Hong Kong,
[6]Knowin AI, [7]Independent Researcher

## ABSTRACT

Large language models (LLMs) are deployed on mobile devices to power killer applications such as intelligent assistants. LLMs pre-trained on general corpora often hallucinate when handling personalized or unseen queries, leading to incorrect or outdated responses. Knowledge editing addresses this by identifying and adjusting a small crucial portion of model weights, without compromising the general knowledge. However, prior knowledge editing methods are impractical to run on local devices due to the resource-heavy backpropagation (BP) needed for updates. We present `MobiEdit`[1], the first mobile knowledge editing framework that enables efficient LLM personalization on commercial off-the-shelf (COTS) mobile devices. `MobiEdit` replaces full-precision BP with quantized forward-only gradient estimation, thus compatible with the energy-efficient mobile neural processing units (NPUs). To further improve gradient estimation efficiency, we introduce two optimizations: an early stopping mechanism that adaptively terminates editing upon success and prefix activation reusing that reduce redundant computation across steps. Our approach enables real-time editing of 3B-parameter models (Qwen2.5-3B-Instruct and Llama3.2-3B-Instruct) on COTS mobile devices with $7.1\times$ less memory, $15.8 \times$ less energy and $3.4\times$ less latency compared to previous knowledge editing methods.

## 1 INTRODUCTION

Mobile LLMs are transitioning from research to real-world deployment, empowering privacy-preserving and latency-sensitive applications such as personal agents (Apple (2024); Li et al. (2024)). While mobile LLMs already embed general world knowledge, a personalized knowledge is crucial for better understanding individual users. This user-specific information is typically absent or diluted during pre-training on large public datasets. As Figure 1 shows, if the user provides the address in one conversation, personalized LLM assistant memorizes the address and applies this information to the future request.

To achieve such personalization, several approaches have been explored in LLMs, including retrieval-augmented generation (RAG) (Fan et al. (2024)), fine-tuning, and knowledge editing (Wang et al. (2024b)). RAG expands prompts by retrieving external knowledge without changing model parameters. However, it adds inference overhead and relies heavily on strong in-context learning capabilities, which are typically weak in small LLMs deployed on mobile devices (Lu et al. (2024)). Fine-tuning requires a large number of training samples for each target fact. It updates model parameters with high computational cost and faces challenges in collecting sufficient data on-device.

In contrast, knowledge editing modifies only a small subset of parameters to inject new knowledge. It maintains high inference speed and can incorporate a single fact with just one training sample. These characteristics make it well-suited for resource-constrained mobile environments. A prevailing knowledge editing paradigm is the locate-and-edit approach (Dai et al. (2022); Meng et al. (2022; 2023); Gu et al. (2024); Fang et al. (2025)), which first identifies influential parameters and

---

[1]The project is released at `https://github.com/UbiquitousLearning/MobiEdit`.

then modifies them by introducing a perturbation optimized through backpropagation to produce the expected output.

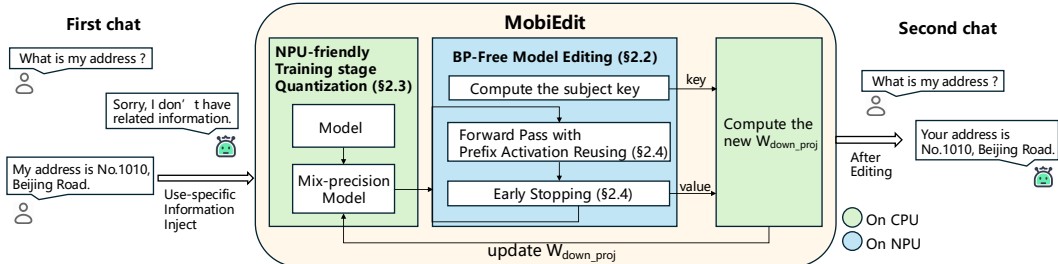

Figure 1: The on-device LLM remembers user information from the first interaction and applies it to subsequent requests.

Despite being effective in updating model knowledge, current knowledge editing methods relying on backpropagation (BP) face three critical challenges for mobile deployment: (1) Incompatibility with forward operators on mobile NPUs. Modern smartphones are equipped with high-performance and energy-efficient NPUs, such as Google edge TPU and Qualcomm Hexagon. These NPUs are designed and optimized primarily for LLM inference, providing up to $60\times$ speedup over CPUs (Xu et al. (2025)). However, training-specific operations are often unsupported or poorly optimized on mobile NPUs (Xu et al. (2024)), rendering BP-based knowledge editing methods infeasible in practice. (2) Poor quantization support. BP-based training operations are often unstable or ineffective on fully quantized models (§2.3). This lack of quantization support exacerbates the previous two challenges: the memory footprint of model parameters remains prohibitively large because full-precision weights must be stored and updated, and it prevents efficient execution on mobile NPUs, which are specifically optimized for low-bit integer computation. (3) Substantial memory overhead. For example, editing a 3B-parameter LLM (Qwen2.5-3B-Instruct) using the classic locate-and-edit method ROME (Meng et al. (2022)) requires over 40 GB of memory to both the full-precision model and activations for BP, which far exceeds the memory capacity of smartphones, typically less than 16 GB.

The above challenges confine current knowledge editing methods to cloud-based computation, undermining two key advantages of mobile LLMs: privacy preservation and offline availability. In this paper, we propose `MobiEdit`, the first mobile knowledge editing system for efficient LLMs personalization. `MobiEdit` is designed to be memory-efficient, NPU-friendly, and compatible with quantization, targeting practical deployment on resource-constrained COTS mobile devices.

**Our solution.** To unleash the power of mobile NPUs, `MobiEdit` builds atop ROME, a widely used locate-and-edit scheme, with a few key building blocks renovated: (1) BP-free training. Instead of calculating standard gradients using BP, `MobiEdit` uses zeroth-order optimization approach to estimate the gradients. `MobiEdit` operates entirely through forward passes, which is memory-efficient and well-suited for mobile NPUs. (2) NPU-friendly training stage quantization. `MobiEdit` introduces a new quantization paradigm for efficient knowledge editing on mobile NPUs. Unlike previous BP-based low-precision training, our forward-only gradient updating is more stable under quantized computation. We thereby quantize all model parameters except the critical projection weights essential to knowledge editing. Only a small portion of weights undergoes full-precision computation to conduct precise gradient estimation. (3) The combination of BP-free training and NPU-friendly quantization further reduces memory consumption. Without the need for backpropagation, activations are not stored, and only INT8-quantized weights are maintained.

We introduce two optimizations to further improve system efficiency: a *prefix activation reusing* that stores KV cache and MLP activations across editing steps, and an *early stopping* that adaptively terminates editing once success criteria are met for different knowledge. Together, these design choices make `MobiEdit` efficient and practical for mobile deployment.

**Results.** We test our method on three COTS mobile phones, Redmi K60 Pro, Redmi K70 Pro and OnePlus 13. On the ZsRE and CounterFact datasets, `MobiEdit` achieves comparable edit

quality while reducing memory usage by $7.1\times$, editing latency by $3.4\times$, and energy consumption by $15.8\times$ compared to different knowledge editing methods, ROME (Meng et al. (2022)), MEMIT (Meng et al. (2023)), WISE (Wang et al. (2024a)), and AlphaEdit (Fang et al. (2025)). To our best knowledge, `MobiEdit` is the first system to make LLM knowledge editing feasible on commercial smartphones.

## 2 METHOD

### 2.1 MOBIEDIT OVERVIEW

Locate-and-edit methods (e.g., ROME, MEMIT) formulate knowledge editing as updating the key–value memory in the MLP layers of LLMs: the subject token is mapped to a *key* $k_*$, and a *value* $v$ vector representing the target fact is optimized, after which the weight matrix is updated so that $Wk_* = v$. This paradigm is effective for accuracy and locality, but existing methods rely on multi-step backpropagation to obtain $v$, which leads to large memory footprint and high latency, rendering them infeasible on forward-only NPUs and resource-constrained mobile devices.

`MobiEdit` inherits this locate-and-edit formulation but present a complete algorithm–hardware co-designed system to enable practical on-device deployment (Figure 1). Specifically, `MobiEdit` performs edits in three coordinated stages:

(1) **NPU-friendly training stage quantization**, converting the original model into a mixed-precision format aligned with mobile NPU constraints;

(2) **BP-Free model editing**, replacing the backpropagation-based optimizer with a forward-only gradient estimator, further enhanced by prefix activation reusing and early stopping;

(3) **Model update**, applying the optimized value vector to modify the mixed-precision model in-place, so that subsequent interactions can immediately benefit from the injected knowledge.

In summary, `MobiEdit` retains the effective key–value editing paradigm while replacing its back-propagation with a lightweight forward-only workflow, augmented by quantization and system-level optimizations for mobile NPUs.

### 2.2 BP-FREE KNOWLEDGE EDITING

Unlike traditional methods that minimize this loss using backpropagation-based gradients, `MobiEdit` estimate the gradients using only forward passes. Specifically, we employ a central-difference estimator along sampled perturbation directions. Given a perturbation direction $u \sim \mathcal{N}(0, I)$, the directional gradient estimate is computed as (Baydin et al. (2022)):

$$\widehat{\nabla}_v \mathcal{L} = \frac{\mathcal{L}(W + \mu u) - \mathcal{L}(W - \mu u)}{2\mu} \cdot u, \tag{1}$$

where $\mu > 0$ is a small scalar step size. To further reduce variance and stabilize training, we average over $N$ independently sampled directions $u_i \sim \mathcal{N}(0, I)$:

$$\widehat{\nabla}_v \mathcal{L} = \frac{1}{N} \sum_{i=1}^{N} \frac{\mathcal{L}(v + \mu u_i) - \mathcal{L}(v - \mu u_i)}{2\mu} \cdot u_i. \tag{2}$$

We then update the value vector as $v \leftarrow v - \eta \widehat{\nabla}_v \mathcal{L}$, where $\eta$ is the learning rate, and repeat until convergence. This procedure performs approximate gradient descent using only forward evaluations. We apply this process iteratively to update $v$. Once the optimal $v^*$ is obtained, we apply a closed-form rank-one weight update:

$$\widehat{W} = W + \Lambda \left(C^{-1} k_*\right)^\top, \quad \text{where} \quad \Lambda = \frac{v^* - W k_*}{(C^{-1} k_*)^\top k_*}. \tag{3}$$

Here $C = KK^\top$ is the estimated key covariance matrix, computed from a sample of key vectors extracted from the model's activation statistics. This update effectively inserts the pair $(k_*, v^*)$ into the MLP's key-value memory, enabling the model to recall the new factual association.

Figure 2: `MobiEdit` quantization workflow and strategy. `MobiEdit` applies quantization to all activations, while retaining floating-point precision only for the editing layer and its preceding layer.

**Benefits of BP-free editing.** BP-free editing is well-suited for mobile NPUs that only support forward passes. In terms of memory efficiency, activations, which take more than 40% of BP-based memory consumption, can be invalidated immediately after the forward pass, because only the final output is required to compute the estimated gradients.

### 2.3 NPU-FRIENDLY TRAINING STAGE QUANTIZATION FOR BP-FREE EDITING

Although activations no longer need to be stored, the large size of LLM weights can still exhaust the memory capacity of mobile devices. A 3B-parameter model, such as Qwen2.5-3B-instruct, requires about 12 GB of full-precision weights. While the newest COTS mobile devices, such as the Xiaomi 15, typically have 16 GB of RAM, only around 75% of this is available for applications, leaving roughly 12 GB usable. As a result, loading the model alone can exhaust available memory and lead to frequent out-of-memory (OOM) errors.

Besides, mobile NPUs are best suited for accelerating INT matrix multiplication with 1024-bit INT8 vector arithmetic. Their floating-point computation capabilities are relatively weak compared to mobile GPUs. To avoid frequent memory swapping, which can severely shorten the lifespan of mobile SSDs, and to leverage the computational advantages of NPUs, we propose an NPU-friendly quantization workflow for BP-free editing.

**Quantization workflow and strategy.** Figure 2 illustrates the quantization workflow of `MobiEdit`. Due to the hardware constraints of mobile NPUs, `MobiEdit` employs a static quantization strategy. The static scales for quantization are determined using representative corpora data. To balance efficiency and accuracy, `MobiEdit` adopts a mixed-precision editing approach: the editing vector and its preceding linear layer are executed in floating-point format; while all other weights are quantized to 8/16-bit integers. This design is informed by two key observations: (i) `MobiEdit` modifies only a small set of parameters proportional to the hidden size. Even minor quantization errors in this context can significantly impact editing accuracy. Furthermore, since the editing module modifies the knowledge stored in its preceding linear layer (Meng et al. (2022)), floating-point precision in this layer is also crucial to maintain accuracy. (ii) The edited vector, being of limited size (equal to the hidden size), results in a negligible computational cost when floating-point precision is used for the editing module and its preceding linear layer. For example, in the Qwen-2.5-3B model, these computations account for only 0.89% of the overall computation, making the performance overhead of using floating-point precision minimal.

**Advantage of our quantization.** `MobiEdit` is more robust to quantization-induced errors than backpropagation (BP)-based editing methods. Consider an $L$-layer Transformer network where all weights and activations are fully quantized:

$$W_\ell^q = W_\ell + \epsilon_{W,\ell}, \quad a_\ell^q = a_\ell + \epsilon_{a,\ell}, \tag{4}$$

where $W_\ell^q$ and $a_\ell^q$ are the quantized weights and activations of the $\ell$-th layer, and $\epsilon_{W,\ell}$ and $\epsilon_{a,\ell}$ denote zero-mean i.i.d. quantization errors with variance $\sigma^2$, independent across layers and forward passes. In each forward pass, quantization noise accumulates recursively.

Without losing generality, we take a linear function $f_\ell(x) = x$ as an intuitive example to analyze the noise accumulation phenomenon, then we have:

$$a_L^q = W_L W_{L-1} \cdots W_1 x + \sum_{j=1}^{L} \Big( \prod_{k=j+1}^{L} W_k \Big) \epsilon_j, \tag{5}$$

where $\epsilon_j$ represents the combined quantization error at layer $j$. Consequently, the total variance of the output noise is

$$\text{Var}[a_L^q - a_L] = \sum_{j=1}^{L} \left\| \prod_{k=j+1}^{L} W_k \right\|^2 \sigma^2 \sim O(L\sigma^2), \quad (6)$$

which grows only linearly with network depth $L$.

Backpropagation further amplifies this noise via the chain rule. For a small edit $\Delta$ in layer $\ell$, the gradient is:

$$g_\ell^{\text{BP}} = \left( \prod_{j=\ell+1}^{L} W_j^\top \right) \frac{\partial \mathcal{L}}{\partial a_L}. \quad (7)$$

Since the weights themselves are quantized, each term in this product carries quantization noise. The variance of the gradient noise therefore becomes

$$\text{Var}[g_\ell^{\text{BP}}] = \sigma^2 \prod_{j=\ell+1}^{L} \|W_j\|^2 \sim O\big(\sigma^2 \cdot \underbrace{\|W\|^{2(L-\ell)}}_{\text{exponential in depth}} \big), \quad (8)$$

indicating that deeper networks lead to an exponential escalation of noise, in sharp contrast to the linear growth in forward propagation.

It is worth noting that in quantization-aware training (QAT) (Jacob et al. (2018)), the Straight-Through Estimator (STE) (Courbariaux et al. (2016)) avoids layer-wise error accumulation by computing gradients on float master weights while treating fake quantization as identity. This is reasonable in QAT, as the goal is to update float weights before re-quantization. In our on-device setting, weights are already quantized, with no float copy or fake quantization; both forward and backward passes operate fully in the quantized domain. Here, quantization errors still accumulate in the forward pass, and BP-based editing further compounds them exponentially.

In contrast to backpropagation, `MobiEdit` only accumulates noise during the forward pass. The zeroth-order gradient is estimated via the centered difference:

$$g_\ell^{\text{ZO}} = \frac{\mathcal{L}_\Delta - \mathcal{L}_{-\Delta}}{2\Delta}, \quad (9)$$

where $\mathcal{L}_\Delta$ and $\mathcal{L}_{-\Delta}$ are the losses obtained from two independent forward passes with perturbed weights $W_\ell^q \pm \Delta$. Each forward pass incurs the same forward-only quantization error derived above, with variance $\text{Var}_{\text{fw}} \sim O(L\sigma^2)$. Consequently, the variance of the ZO estimator is

$$\text{Var}[g_\ell^{\text{ZO}}] = \frac{\text{Var}[\mathcal{L}_\Delta] + \text{Var}[\mathcal{L}_{-\Delta}]}{(2\Delta)^2} = \frac{2\,\text{Var}_{\text{fw}}}{4\Delta^2} \sim O\big(L\sigma^2/\Delta^2\big), \quad (10)$$

which grows only linearly in depth $L$, in sharp contrast to the exponential dependence of backpropagation and explaining why `MobiEdit` is more robust in deep quantized networks.

**Quantization robustness evaluation of BP-free training.** Our experiments in Table 1 demonstrates the fragility of BP-based editing under quantization and the robustness of zeroth-order optimizations. ROME's editing success rate drops sharply from 96% in the FP32 setting to 41% in the W8A16 setting due to high gradient noise in low-bit conditions. In contrast, `MobiEdit` achieves a score of 80 under the same W8A16 quantization, demonstrating substantially higher robustness.

## 2.4 FURTHER OPTIMIZATIONS

Although a single step of `MobiEdit` is efficient on mobile NPUs, a remaining challenge is that it requires significantly more steps to reach comparable convergence performance. Each gradient estimate in zeroth-order optimization is taken along a single perturbation direction, causing deviation from the true gradient and requiring many more steps to reach stable convergence. For example, `MobiEdit` requires on average $20\times$ more steps than BP-based model editing on the ZsRE and CounterFact datasets, which eliminates its efficiency advantage in terms of wall-clock time.

Table 1: Impact of quantization on edit success for ROME (BP) and `MobiEdit` (BP-Free).

| Method | Precision | Edit Success Score |
|---|---|---|
| ROME (BP) | FP32 | 96 |
| ROME (BP) | W8A16 | 41 |
| MobiEdit (BP-Free) | FP32 | 86 |
| MobiEdit (BP-Free) | W8A16 | 80 |

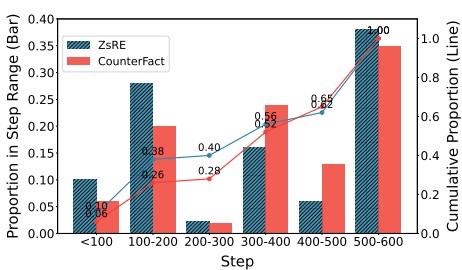

Figure 3: Distribution of editing steps for successful edits.

**Early stopping.** To address this, we first analyze the successful editing step distribution of edited facts. As shown in Figure 3, we find that different knowledge has different editing difficulty. Based on this observation, we introduce a lightweight early stopping mechanism that adaptively determines the editing horizon based on runtime success feedback. Specifically, during editing, we periodically evaluate the model's response to the edited fact every $M$ steps (e.g., every 20 steps). The editing process is terminated early once the model satisfies a pre-defined success criterion. This typically occurs when it produces the desired target output with a confidence above a given threshold $m$. The early stopping mechanism automatically adjusts the editing steps to the complexity of each edit instance, avoiding unnecessary forward passes for easy-to-edit facts and reducing overfitting risk by stopping at the point of first success.

**Prefix activation reusing.** In our knowledge editing setting, each optimization step uses the same set of input examples constructed by combining a fixed set of randomly sampled prefixes with the fact to be edited. Formally, for a target fact $f$, we define a set of editing inputs as:

$$\mathcal{X}_{\text{edit}} = \{[p_1 + f], [p_2 + f], \ldots, [p_n + f]\}, \tag{11}$$

where $p_1, p_2, \ldots, p_n$ are different randomly sampled prefixes.

We observe that in each step, the prefix tokens in the input do not change, and therefore their corresponding activations are recomputed redundantly. To reduce this overhead, we introduce *Prefix activation reusing*: during the first step, we cache the intermediate activations for the prefix tokens, and in all subsequent steps, these cached activations are inherited across editing steps without recomputation, while only the fact tokens are recomputed. This greatly reduces compute without changing the model architecture or input format.

Reusing "stale" activations from previous steps introduces an accuracy-efficiency trade-off. We observe this effect in Figure 5, which visualizes the "staleness" by showing the cosine similarity of activations between consecutive steps. While the similarity is high in many layers, it gradually decreases in others, especially as editing progresses, indicating that the cached prefix representations are not perfectly aligned with the evolving model. The full implications of this trade-off, particularly under quantization, are analyzed in detail in Section 3.3.

To avoid the staleness accumulates from halting the editing process, we re-compute the prefix activations as long as the editing loss does not decrease by 0.001 over 3 steps.

## 3 EXPERIMENTS

### 3.1 SETUP

**Baselines.** We compare `MobiEdit` against four representative locate-and-edit methods: ROME (Meng et al. (2022)), MEMIT (Meng et al. (2023)), AlphaEdit (Fang et al. (2025)), and WISE (Wang et al. (2024a)). These methods follow the same paradigm of identifying key activations and injecting new knowledge into MLP layers, but differ in target granularity and update mechanisms. ROME performs single-layer editing. MEMIT extends it to multi-fact scenarios. Al-

phaEdit uses null-space projections for preservation, and WISE incorporates dynamic routing to FFNs that store facts.

**Datasets and model.** We evaluate `MobiEdit` on two standard datasets widely used in factual knowledge editing: ZsRE (Levy et al. (2017)), a zero-shot relation extraction dataset derived from WikiRE, and CounterFact (Meng et al. (2022)), a curated benchmark of factual edits targeting named entities (e.g., people, locations), with truth and counterfactual contexts. These two datasets jointly assess *edit success* (correctly producing the target knowledge), *locality* (preserving unrelated knowledge), and *portability* (applying the edited knowledge across contexts), the three key metrics for knowledge editing. We use Qwen2.5-3B-Instruct (QwenTeam (2024)) and Llama3.2-3B-Instruct (AI@Meta (2024)) as our target models.

Table 2: Devices used in experiments.

| Device | SoC | RAM | NPU |
|---|---|---|---|
| Redmi K60 Pro | Snapdragon 8 Gen 2 | 16GB LPDDR5 | Hexagon NPU V73 |
| Redmi K70 Pro | Snapdragon 8 Gen 3 | 16GB LPDDR5 | Hexagon NPU V75 |
| OnePlus 13 | Snapdragon 8 Elite | 24GB LPDDR5 | Hexagon NPU V79 |

**Implementation details.** To assess on-device feasibility, we run all editing procedures on three COTS mobile devices, as shown in Table 2. We perform all experiments using local inference engine mllm-npu (Xu et al. (2025)) optimized for NPU execution. The latency on CPU is obtained by running on mobile phones using llm.c (karpathy (2013)). We use memory swapping while reaching the memory limit. `MobiEdit` uses W8A16 quantization (INT8 weights, INT16 activations), a format widely supported by mobile NPUs and inference engines, ensuring compatibility and throughput. The metric of memory usage in this paper is defined as the total memory required, assuming sufficient memory is available. For a simple comparison, the system efficiency values are first normalized to the range $[40, 100]$ using min-max normalization, and then inverted.

## 3.2 END-TO-END PERFORMANCE

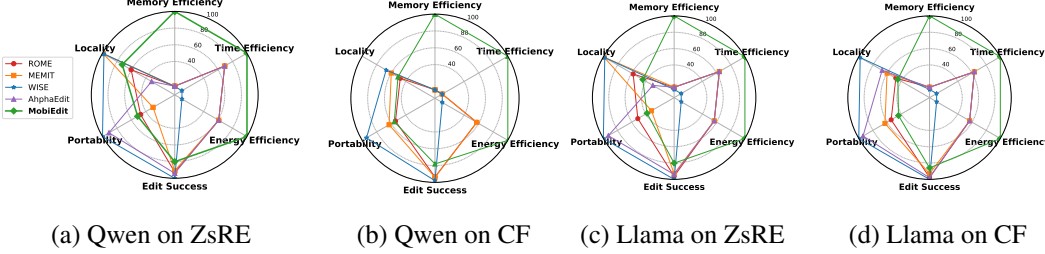

| (a) Qwen on ZsRE | (b) Qwen on CF | (c) Llama on ZsRE | (d) Llama on CF |
|---|---|---|---|

Figure 4: The comprehensive editing ability and performance comparison of knowledge editing methods on the ZsRE and CounterFact datasets.

**Overall results.** We compare `MobiEdit` against all baselines across six dimensions: edit success, locality, portability, time efficiency, memory efficiency, and energy efficiency, as shown in Figure 4. The results showing that **`MobiEdit` achieves a balance of high accuracy and low resource cost.**

On Qwen2.5-3B-Instruct model, `MobiEdit` achieves an 80.1 edit success score, 72.6 locality score, and 51.4 portability score while requiring only 25.5 minutes, 0.018 kJ energy, and 6.2 GB memory. On Llama3.2-3B-Instruct model, `MobiEdit` achieves an edit success score of 88.3, locality score of 44.3, and portability score of 38.3, while requiring only 23.1 minutes, 0.017 kJ energy, and 5.0 GB memory. Although slightly behind in quality, `MobiEdit` significantly outperforms in efficiency, reducing memory usage by more than $7\times$ and energy consumption by over $10\times$. These substantial resource savings make `MobiEdit` the only realistically viable solution for mobile devices.

The significant performance improvement is attributed to `MobiEdit` leveraging a BP-free editing method, which reduces the memory and computational overhead caused by backpropagation. Additionally, `MobiEdit` incorporates mobile hardware-friendly quantization and two optimizations

specifically designed for the bp-free training step. These enhancements not only maximize the performance potential of mobile NPUs but also minimize the number of training steps and eliminate redundant computations.

With `MobiEdit`'s exceptional performance gains, although our method has a drop in edit accuracy, it significantly improves the number of successful edits per unit time, which is critical for real-world usability. Since knowledge editing occupies device resources and cannot be performed while the user is actively using their phone, in practice there are only about 8 hours per day (e.g., when the user is asleep) available for editing. As shown in Table 3, in 8 hours, our method can successfully edit 14 facts, whereas traditional methods only accomplish 2–5. For simple facts, our method requires only 2–3 minutes, while traditional methods incur a fixed processing time of either 1.25 hours or 3 hours for all facts, depending on the specific method used. Running traditional methods also causes a temperature increase of approximately 10 °C after just 100 seconds of sustained CPU load, resulting in device lag or shutdown.

Table 3: Facts edited successfully within 8 hours on Redmi K60 Pro.

| Method | Count |
|---|---|
| ROME | 5 |
| MEMIT | 5 |
| WISE | 2 |
| AlphaEdit | 4 |
| **MobiEdit** | 14 |

**Editing performance.** Table 4 provides a detailed comparison of memory, latency, and energy usage across three commercial smartphones on ZsRE dataset. All baseline methods considered in our experiments, including ROME, MEMIT, AlphaEdit, and WISE, require more than 30 GB of memory. This is due to the lack of memory optimization in llm.c for training part parameters. And their per-edit energy is ranging from 0.18J to 0.36J. For instance, WISE consumes 0.36J and takes 11,359 seconds on K60 for a single edit. Such workloads not only exceed memory budgets but impose intense thermal and scheduling pressure on mobile hardware.

• *Qwen2.5-3B-Instruct.* `MobiEdit` consumes only 6.2GB memory under 0.03J energy per edit across all devices, completing edits in 1211 to 1902 seconds. This 10× energy reduction allows `MobiEdit` to run editing workloads unobtrusively in the background without interrupting the user experience or triggering thermal limits. Such sustainability is critical for real-world mobile applications, where knowledge editing may be triggered interactively under tight system constraints.

• *Llama3.2-3B-Instruct.* `MobiEdit` uses 5 GB, achieving 3.1× to 6.1× lower latency and 13.6× to 27.6× higher energy efficiency than baselines. The average latency is 1411 seconds on Llama3.2-3B compared with 1530 seconds on Qwen2.5-3B, since Llama3.2 adopts a wider and shallower architecture (3072×28 vs. 2048×36), being more computationally favorable for NPUs.

Appendix A and B details that the performance gains come from our algorithm–hardware co-design: (i) eliminating backward passes, (ii) using prefix activation reusing to cut redundant computation, and (iii) aligning computation precision with NPU characteristics.

## 3.3 Ablation Study

On Llama3.2-8B-Instruct, `MobiEdit` achieves an 85.3 edit success score. Our ablation study indicates that prefix activation reusing is the main source of the quality trade-off, reducing the success score by 8.4 points, while quantization has a much milder impact, with only a 2.1 point drop.

To understand why prefix reusing has such a significant impact, we provide a detailed visual analysis contrasting the activation dynamics in FP32 and quantized settings. First, in an ideal FP32 setting as Figure 5a shown, we observed that activations are extremely stable, which provided the core motivation for our prefix reusing optimization. As shown below, the similarity between consecutive steps is very high ($>0.90$) for most layers. However, a slight decrease in similarity can still be seen in the top and bottom layers over time, illustrating the "staleness" effect.

Second, we found that quantization significantly amplifies this instability. Figure 5b shows the analysis in our practical W8A16 quantization with INT8 weights and INT16 activations. Several layers now exhibit consistently low similarity (the dark horizontal bands), indicating significant volatility. Interestingly, these volatile layers are concentrated near the input and output of the network. This aligns with observations in the quantization literature(An et al. (2025)), which find that outlier features, a key challenge for quantization, are most prominent in the first and last Transformer layers. Quantizing tensors with large outlier values requires a large quantization step size. Due to this large step size, a small perturbation to a pre-quantized activation value can easily cause it to be

Table 4: Performance comparison of our method with NPU and other knowledge editing methods with CPU on different devices.

(a) Qwen2.5-3B-Instruct

| Method | Memory (GB) | K60 | | K70 | | OnePlus | |
|---|---|---|---|---|---|---|---|
| | | Time (s) | Energy (kJ) | Time (s) | Energy (kJ) | Time (s) | Energy (kJ) |
| ROME | 46.14 | 4543.78 | 25.13 | 4276.49 | 24.23 | 3252.81 | 18.02 |
| MEMIT | 46.14 | 4543.78 | 25.13 | 4276.49 | 24.23 | 3252.81 | 18.02 |
| WISE | 46.30 | 11359.44 | 63.82 | 8552.99 | 47.16 | 6505.63 | 36.05 |
| AlphaEdit | 46.14 | 4543.78 | 25.13 | 4276.49 | 24.23 | 3252.81 | 18.02 |
| **MobiEdit** | **6.20** | **1902.88** | **0.023** | **1477.67** | **0.018** | **1211.83** | **0.014** |

(b) Llama3.2-3B-Instruct

| Method | Memory (GB) | K60 | | K70 | | OnePlus | |
|---|---|---|---|---|---|---|---|
| | | Time (s) | Energy (kJ) | Time (s) | Energy (kJ) | Time (s) | Energy (kJ) |
| ROME | 34.14 | 4834.78 | 27.01 | 4578.66 | 25.23 | 3551.82 | 20.22 |
| MEMIT | 34.14 | 4834.78 | 27.01 | 4578.66 | 25.23 | 3551.82 | 20.22 |
| WISE | 35.05 | 9668.86 | 53.89 | 9157.32 | 50.45 | 6505.63 | 39.32 |
| AlphaEdit | 34.14 | 4834.78 | 27.01 | 4578.66 | 25.23 | 3551.82 | 20.22 |
| **MobiEdit** | **5.06** | **1754.26** | **2.10** | **1362.26** | **1.71** | **1117.19** | **1.44** |

Table 5: Edit Quality Comparison for Llama3.2-8B-Instruct on ZsRE dataset.

| | ROME | zo | zo+prefix | zo+prefix+quan |
|---|---|---|---|---|
| Edit Succ | 96.2 | 95.8 | 87.4 | 85.3 |
| Portability | 51.3 | 50.6 | 33.2 | 38.3 |
| Locality | 57.9 | 66.6 | 56.2 | 66.3 |
| Fluency | 555 | 580 | 566 | 582 |

mapped to a different integer after quantization.This results in a large, discrete change in the final quantized activation, even though the initial perturbation was small and continuous. This effect is most pronounced in outlier-rich layers and explains the significant drop in activation similarity we observe.

This comparative analysis directly explains the source of the quality trade-off: it stems from a compounding effect. Our optimization reuses activations that are not only stale but also noisy and imprecise due to quantization. The interaction of these two factors leads to less accurate gradient estimates. To be specific on why Portability and Locality are hurt more: These two metrics require enough precision. Portability needs to update a semantic space for generalization. Locality needs to isolate the edit to avoid damaging unrelated facts. The gradient estimated from stale, quantized activations is slightly blurry. While it is accurate enough to ensure Edit Success, it lacks the surgical precision to perfectly remap the semantic space and prevent leaks into unrelated knowledge.

## 4 RELATED WORK

**Knowledge editing.** Existing methods include *locate-and-edit* (ROME (Meng et al. (2022)), MEMIT (Meng et al. (2023)), AlphaEdit (Fang et al. (2025)), WISE (Wang et al. (2024a))), which directly modify model weights; *retrieval-augmented* (e.g., RECIPE (Chen et al. (2024))), which fetch external knowledge; and *meta-learning* (MEND (Mitchell et al. (2021))), which learns editing strategies from examples. Li et al. (2025) improves editing algorithms in unlearning settings. While having good editing quality, these approaches rely on multi-step backpropagation, leading to high memory and latency. We instead target memory-efficient, NPU-friendly, quantization-compatible editing.

**BP-Free training.** Zeroth-order methods such as FwdLLM (Xu et al. (2024)) and MeZO (Malladi et al. (2023)) estimate gradients via forward-pass perturbations and enable high-quality adap-

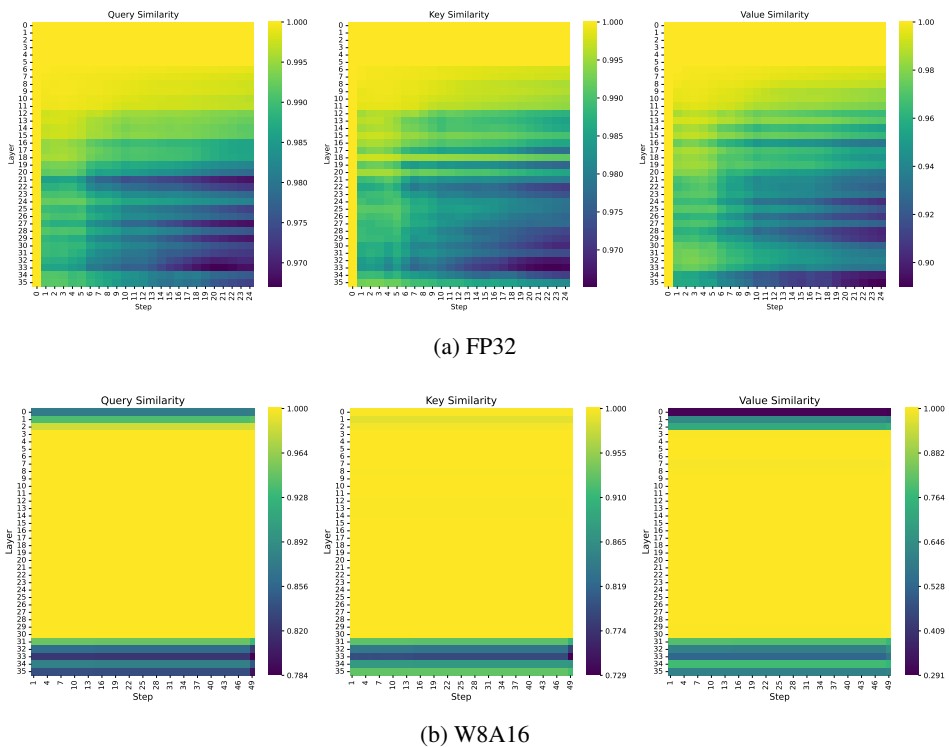

Figure 5: Stability of inherited prefix activations over editing steps. The heatmap shows the cosine similarity between the QKV activations at each step and those cached at step 0, demonstrating that the inherited activations remain a close approximation even after multiple updates.

tation, but are designed for downstream task tuning, often requiring thousands of steps, and have not been applied to factual editing. To our knowledge, no prior work supports robust fact injection under both forward-only and quantized constraints. Our framework addresses this gap with BP-free, quantization-aware editing for edge deployment.

**Quantization.** By representing weights and activations in low-bit formats, quantization reduces the memory and computational cost of LLMs. SmoothQuant (Xiao et al. (2023)) and Spin-Quant (Yuan et al. (2025)) mitigate activation/weight outliers, enabling accurate low-bit models, while QLoRA (Dettmers et al. (2023)) combines 4-bit quantization with low-rank adapters for memory-efficient fine-tuning. Unlike these inference- or fine-tuning-oriented approaches, we integrate mixed-precision quantization with BP-free editing to update knowledge directly on NPUs.

## 5 CONCLUSION

We present `MobiEdit`, the first mobile-compatible framework for efficient knowledge editing in large language models. It uses quantized, forward-only gradient estimation to meet the compute and memory limits of commercial NPUs, and applies prefix activation reusing and early stopping for further speedup. `MobiEdit` enables real-time, on-device editing of 3B models, reducing memory by $7.1\times$, energy by $15.8\times$, and latency by 72% over prior methods, making it a practical solution for user-driven LLM personalization on mobile and edge devices.

## ACKNOWLEDGMENTS

We thank the anonymous reviewers for their feedback on this work. We extend our gratitude to Prof. Nicholas D. Lane for insightful discussions and helpful feedback that greatly improved this work. This research is funded by the Beijing Municipal Science & Technology Commission, the Administrative Commission of Zhongguancun Science Park, NSFC (No. 62522202), Beijing Natural Science Foundation (No. L253005), the Shenzhen Science and Technology Program with Grant No. JCYJ20241202124021028, the Major Key Project of PCL under Grant PCL2025A10 and PCL2024A06, and in part by the Shenzhen Science and Technology Program under Grant RCJC20231211085918010.

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

# A  ABLATION STUDIES ON RUNTIME COST

## A.1  ABLATION STUDY OF PREFIX ACTIVATION REUSING AND EARLY STOPPING

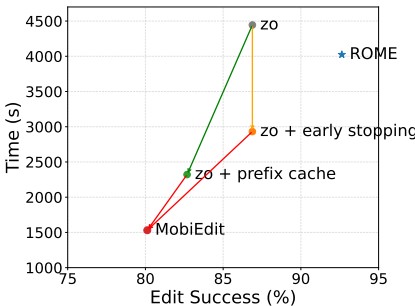

Figure 6: Edit success vs. time on ZsRE. Time is averaged across all devices.

Figure 6 presents an ablation study of the key algorithmic and system-level optimizations in our framework. The basic zeroth-order method (zo) achieves moderate edit success but incurs excessive time cost, often over 4000 seconds per edit. Introducing early stopping (dynamic step controller) alone reduces average editing time by over 40%, without sacrificing accuracy. The early stopping module effectively eliminates redundant optimization once the target knowledge has been successfully fitted. Adding prefix activation reusing further accelerates editing by another 20–30%, as observed across all devices. For each fact, the reusing reduces computation proportionally to the ratio of the length to the total input length. `MobiEdit`, which incorporates both optimizations, reduces editing time to nearly one-third of the baseline zo and achieves the best balance of edit success and efficiency.

## A.2  QUANTIZATION VS. NON-QUANTIZATION IN ROME

We compared `MobiEdit` and ROME (quantized and non-quantized) on CPU and NPU as Table 6. Quantization gives little or even negative benefit for BP-based methods on CPU. Editing time for ROME changes from 123.7s to 127.1s. Without NPU utilization, `MobiEdit` running on the Apple CPU shows no speedup compared to ROME. Therefore, our main acceleration is observed only for `MobiEdit` on NPU, driven by algorithm–hardware co-design. These results highlight the necessity and complementarity of each system-level optimization in achieving practical on-device knowledge editing, delivering substantial improvements in both hardware and algorithmic efficiency.

## A.3  COMPARISON WITH OTHER CPU BACKEND FRAMEWORK

### A.3.1  COMPARISON WITH A SOTA ON-DEVICE FINETUNING METHOD

To provide a fair comparison against a state-of-the-art, gradient-based method, we compare `MobiEdit` with XPerT (Wang et al. (2025)), a personalized language style fine-tuning framework on device. We compare our power consumption with XPerT Wang et al. (2025) while editing a

Table 6: Editing efficiency of ROME and `MobiEdit` on Apple CPU, K60 CPU and Hexagon NPU V73. Quantization alone (ROME W8A16) yields no significant speedup. The major performance gains come from NPU utilization combined with hardware–software co-design.

| Method | HW | Precision | Forward/Step | Backward/Step | Steps/Edit | Total/Edit (s) |
|---|---|---|---|---|---|---|
| ROME | Apple CPU | FP32 | 1121.3 ms | 5065.1 ms | 20 | 123.7 |
| ROME | Apple CPU | W8A16 | 1048.3 ms | 5305.7 ms | 20 | 127.1 |
| MobiEdit | Apple CPU | W8A16 | 429 ms | N/A | $10 \times 397$ | 1705.8 |
| ROME | K60 CPU | FP32 | 38 s | 182 s | 20 | 5100 |
| MobiEdit | NPU | W8A16 | 331 ms | N/A | $10 \times 397$ | 1746 |

Llama3.2-1B model on a Snapdragon 8 Gen 2 device. As Table 7 shows, Our MobiEdit achieves a $1.7\times$ speedup and is $10\times$ more power-efficient than the BP-based XPerT under the same hardware and model conditions.

Table 7: Power Consumption Comparison with XPerT (Wang et al. (2025))

| Method | Time (s) | Power (W) |
|---|---|---|
| XperT (CPU) | 1074 | 2.6 |
| MobiEdit (NPU) | 634 | 0.27 |

### A.3.2 COMPARISON WITH A `LLAMA.CPP`-BASED FORWARD-ONLY EDITING METHOD

Second, due to our method's forward-only feature, we can leverage `llama.cpp` to create a highly-optimized CPU baseline for our own algorithm. We implemented this as `llama.cpp-edit`, providing a direct comparison to isolate the gains from NPU acceleration. The comparison is conducted on a Redmi K70 device using the Qwen2.5-3B model quantized to INT4. For the CPU backend, `llama.cpp` was configured to run on 8 cores and 8 threads. Leveraging the NPU provides an additional $2\times$ speedup and $5\times$ higher energy efficiency over a top-tier CPU framework for the same editing task. Together, these analyses prove our efficiency gains from a hardware-aware

Table 8: `MobiEdit` performance on Optimized CPU Backend versus on NPU

| Method | Time (s) | Power (W) |
|---|---|---|
| llama.cpp-edit (CPU) | 2423.16 | 4.33 |
| MobiEdit (NPU) | 1222.61 | 0.83 |

co-design that unlocks the NPU's raw power.

## B THE PERFORMANCE BREAKDOWN ANALYSIS

### B.1 COMPUTATION BREAKDOWN

To clarify the time efficiency of `MobiEdit`, we provide a detailed breakdown for Qwen2.5-3B-Instruct on K60. Since most sequences are between 16 and 32 tokens in ZsRE dataset, we take a sequence length of 24 (with 10 tokens as ) and batch size of 7 (7 inputs with different ) for illustration. For BP-based editing methods (such as ROME): The editing time is about 5100s. On CPU, a single forward pass takes 38s and a single backward pass takes 182s. Each edit requires 20 steps. For `MobiEdit` on NPU: One knowledge edit takes approximately 1746s. A single NPU forward pass takes 752 ms. For each optimization step, we use 5 perturbations, so one step takes 7.5s. Since the is 10 tokens out of the 24, prefix activation reusing allows us to save 41% computation. One step takes about 4.4s. According to our analysis on the ZsRE dataset, the average number of steps to edit one fact is 397 step. On the example, `MobiEdit` achieves a speedup of about 2.92x compared to BP-based editing and match the data in Figure 5.

## B.2 Implementation details and Bottleneck Analysis

We analyze with implementation details regarding computation and memory access during forward passes of Qwen2.5-3B-Instruction on the K60 NPU. In our implementation, we effectively treat the sequence length as 1680, since each sequence contains 24 tokens and we process 7 different sequences simultaneously. Additionally, we generate 10 distinct edit vectors from 5 different perturbations, all of which need to be processed during the forward pass. These edit vectors (each of size 1×hidden dimension) are treated as model inputs rather than parameters. This results in a total input length of 24×7×10=1680 tokens per training step. Table 9 presents the architectural hyperparameters of Qwen2.5-3B-Instruction, while Table 10 details the computational capabilities and memory bandwidth of our NPU. Based on these specifications, Table 11 provides a breakdown of the computational workload and memory access requirements for a layer in a single forward pass. Our analysis reveals that most stages are compute bound rather than memory bound. In practice, the NPU's 8MB VTCM cache offers significantly higher bandwidth than the LP-DDR5 memory. The cache enables more efficient access to activation values. Many activations can be served directly from this high-speed cache without accessing main memory, meaning the actual memory access times are substantially faster than our theoretical calculations would suggest. Therefore, we can conclude that the prefix activation reusing delivers near-linear acceleration, significantly enhancing the overall computational efficiency of our implementation.

Table 9: Qwen2.5-3B-Instruct architecture.

| Hyperparameter | Value |
| --- | --- |
| Hidden Size | 2048 |
| Q Head | 16 |
| KV Head | 2 |
| MLP Size | 11008 |
| Layers | 36 |

Table 10: Hardware capability of Xiaomi K60 NPU.

| Attribute | Value |
| --- | --- |
| NPU TOPS | 52 |
| LP-DDR5 Bandwidth (GB/s) | 40 |

Table 11: Computation and memory access analysis per forward step.

| Pipeline | Q Gen | KV Gen | QK | PV | O | MLP Up | MLP Gate | MLP Down |
| --- | --- | --- | --- | --- | --- | --- | --- | --- |
| Computation (GOPs) | 6.563 | 0.820 | 5.383 | 5.383 | 6.563 | 35.273 | 35.273 | 35.273 |
| Memory Traffic (GB) | 0.007 | 0.002 | 0.044 | 0.024 | 0.007 | 0.029 | 0.029 | 0.022 |
| Compute Time (ms) | 0.136 | 0.017 | 0.111 | 0.111 | 0.136 | 0.728 | 0.728 | 0.728 |
| Memory Access Time (ms) | 0.089 | 0.046 | 0.045 | 0.531 | 0.089 | 0.303 | 0.303 | 0.478 |
| Bound | Compute | Memory | Compute | Memory | Compute | Compute | Compute | Compute |

These results highlight that the performance advantage of `MobiEdit` comes from (i) eliminating backward passes, (ii) leveraging prefix activation reusing to reduce redundant computation, and (iii) aligning with NPU compute characteristics.

## C Loss Curve

We illustrate the loss curves under three configurations: ZO, ZO+prefix activation reusing , and ZO+prefix activation reusing+Quantization in Figure 7. With early stopping, all knowledge edits terminated immediately upon successful editing. Notably, prefix activation reusing demonstrated remarkable optimization acceleration. Fact 5 exhibited a 5× reduction in required training steps (from 300 to 60 steps). Although quantization introduced transient oscillations during gradient descent, it maintained final convergence accuracy, albeit with increased optimization time, as demonstrated by Fact 4's step count rising from 40 to 120.

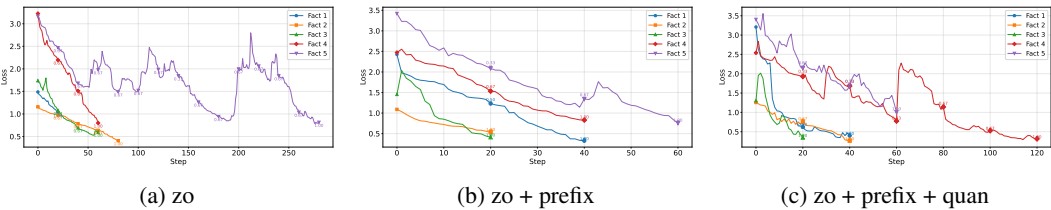

Figure 7: Loss curve of Llama3.2 with ZsRE

# D  HYPERPARAMETERS

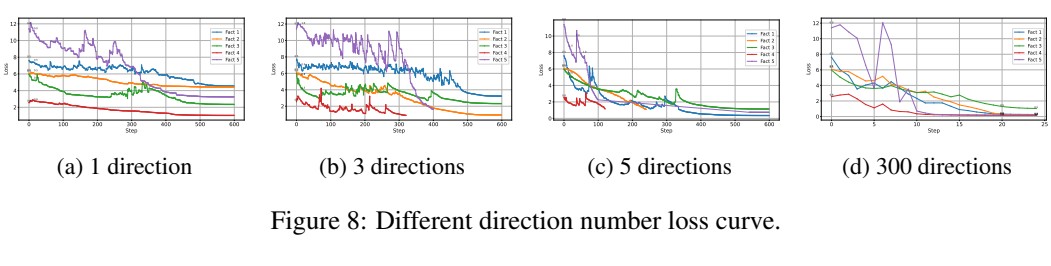

Figure 8: Different direction number loss curve.

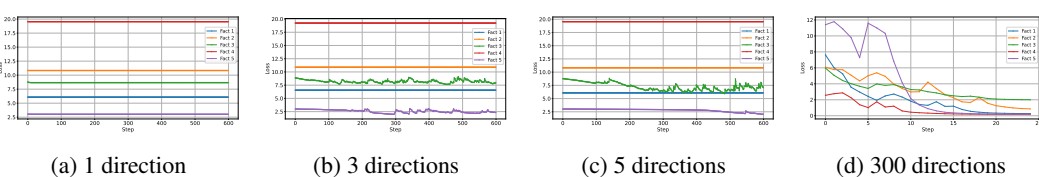

Figure 9: Different direction number loss curve without Cosine Annealing Learning Rate.

We explore the hyperparameters for `MobiEdit`. Figure 8 and Figure 9 analyze the interaction between the number of sampled directions and learning rate schedules in knowledge editing. While prior work on knowledge editing and zeroth-order optimization primarily employed static learning rates, as shown in Figure 8, our experiments highlight the significant advantage of cosine annealing, demonstrated in Figure 9. With a static learning rate, effective parameter updates require at least 300 sampled directions, and optimization completely fails with only 1–5 directions due to persistent loss plateaus. In contrast, cosine annealing achieves noticeable loss reduction with just one sampled direction and delivers practical editing performance with only five directions—a 60× improvement in sampling efficiency. This empirically confirms that adaptive learning rate scheduling inherently reduces gradient estimation noise in zeroth-order optimization, which is especially critical in low-sample regimes where static learning rates face inherent limitations.

# E  EXAMPLES OF PERSONAL INFORMATION INJECTION ON MOBILE

To validate MobiEdit's effectiveness on-device personalization, we have conducted new experiments on a personalization dataset. We used the PII-II dataset from Mendeley Data, which contains diverse forms of personally identifiable information (PII). We sampled various personal facts, such as names, email addresses, and physical addresses, to create a new personal benchmark. MobiEdit demonstrates strong performance on this new benchmark, successfully injecting personal facts into the LLM while maintaining its significant efficiency advantages. The Table 12 below summarizes the results for representative examples. Furthermore, to provide more insight into the editing process, we show several loss curves for these personalization edits in Figure 10. The successful editing of personal information requires nearly 600 steps, which confirms its higher difficulty compared to factual editing. Nevertheless, our method can still accomplish this task successfully in under 50 minutes.

Table 12: MobiEdit Performance on the Personal Information Dataset

| Question | Before Editing | After Editing | Time (s) | Energy (kJ) |
|---|---|---|---|---|
| What is the email address of Andrew Yoder? | I Yoder@us.com | AndrewYoder@company.com | 2739.96 | 2.71 |
| What is the phone number of Andrew Yoder? | Andrew1 719 523-4567 | +1 555-123-4567 | 2909.23 | 2.75 |
| What is the URL associated with Andrew Yoder? | Andrew://en.linkedin.com | https://www.AndrewYoder.com | 3011.68 | 2.78 |

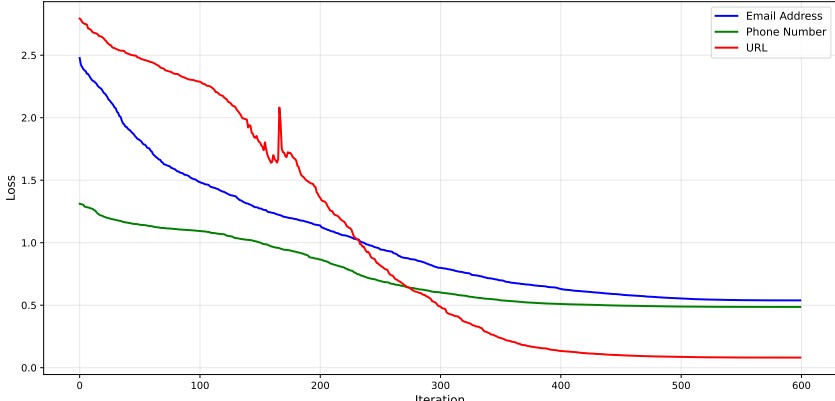

Figure 10: Training loss curves on the Personal Information dataset.

Table 13: MobiEdit Performance on models with different size

| Model | MobiEdit Success Score | ROME Success Score | MobiEdit Time (s) | ROME Time (s) | Speedup |
|---|---|---|---|---|---|
| 1B | 72 | 94 | 634.87 | 1228.22 | 1.93 |
| 3B | 80 | 95 | 1754.60 | 4834.78 | 2.76 |
| 8B | 85 | 94 | 4478.93 | 16982.31 | 3.79 |

## F    MOBIEDIT PERFORMANCE ON DIFFERENT MODEL SIZE

We evaluate MobiEdit on Llama3.2 models of different sizes: 1B, 3B, and 8B. As shown in Table 13, on the 1B model, MobiEdit reaches a success score of 72.0 and gives a 1.93× speedup over ROME, with time cost of 634.87 seconds and 1228.22 seconds. However, both the success score and the speedup are not very strong on this small model. Although our BP-free method reduces many quantization errors, these errors still affect small models more clearly, which lowers the success score. Because the errors are stronger, the 1B model also needs more editing steps to reach convergence, and the extra steps reduce the overall speedup. As the model becomes larger, MobiEdit performs better. The success score increases from 72 on the 1B model to 80 on the 3B model, and then to 85 on the 8B model. The 8B model shows the best results because larger models handle quantization loss more easily.

## G    USE OF LLMS

This work has benefited from the assistance of large language models (LLMs). Specifically, LLMs were used for (i) language proofreading of the manuscript and (ii) generating code to assist in figure creation. The authors critically reviewed and edited all LLM-generated content before use. We are grateful for the availability of these tools, which aided in efficiency without affecting the integrity of the research.

