# OpenReview forum: "MobiEdit: Resource-efficient Knowledge Editing for Personalized On-device LLMs"
_ICLR.cc/2026/Conference — ICLR 2026 Poster_

### Official Review · Reviewer_V7a3 · 2025-10-25

**Soundness:** 3
**Presentation:** 3
**Contribution:** 3
**Rating:** 6
**Confidence:** 3

**Summary:**

This paper presents MobiEdit, the first practical framework for performing knowledge editing (KE) on large language models directly on commercial off-the-shelf (COTS) mobile devices. The work identifies that existing KE methods, which rely on backpropagation (BP), are fundamentally incompatible with mobile deployment due to (1) high memory/energy costs from storing activations, (2) instability when combined with quantization, and (3) incompatibility with mobile NPUs, which are typically forward-pass-only accelerators. MobiEdit proposed BP-Free Editing, NPU-Friendly Quantization, and System Optimizations to solve the challenges. Experiments on COTS mobile phones (e.g., Redmi K60 Pro) show that MobiEdit can edit 3B LLMs (Qwen2.5, Llama3.2) with 7.1x less memory, 15.8x less energy, and 3.4x less latency than BP-based methods, making on-device personalization feasible for the first time.

**Strengths:**

The paper tackles a critical bottleneck for the next generation of on-device personal AI: enabling models to learn from user interactions without resorting to the cloud. To our knowledge, this is the first work to demonstrate practical knowledge editing on commercial mobile NPUs. The "Facts edited successfully within 8 hours" metric (Table 3) is a killer demonstration of this practical feasibility, showing MobiEdit can edit 14 facts while ROME can only edit 2, all while avoiding device overheating. A major strength is the rigorous justification for using zeroth-order (ZO) optimization. The paper provides a clear theoretical analysis (Sec 2.3, Eq 4-9) arguing that BP-based gradient noise accumulates multiplicatively with network depth under quantization, while ZO-based noise remains constant.

**Weaknesses:**

The main results (Fig 5) clearly show a trade-off: MobiEdit has slightly lower "Edit Success," "Portability," and "Locality" than the (infeasible) FP32 ROME baseline. The main text does not fully analyze the source of this drop. Appendix A (Table 5) contains the crucial insight: the "prefix activation reusing" optimization is the main culprit, causing an 8.4-point drop in success, while quantization itself causes only a 2.1-point drop. This is a critical finding and must be discussed in the main paper, as it's the central trade-off of the work (speed vs. accuracy). The claim in Sec 2.4 that reusing stale activations "does not negatively affect the editing outcome" is directly contradicted by Appendix A.

**Questions:**

The main trade-off in the paper is efficiency vs. quality. Your appendix (Table 5) brilliantly isolates the source of this quality drop (mostly from prefix reusing, not quantization). Could you please move this critical analysis into the main paper (e.g., in Sec 3.2 or 2.4) and discuss this trade-off more explicitly? Why do you think reusing stale prefix activations hurts portability and locality so much?

---

> ### Author Response · Authors · 2025-11-21
>
> We are extremely grateful to the reviewer for this insightful and highly constructive feedback. **We have made two major updates to the revised paper based on this excellent advice: (1) We have moved the core ablation study into the main experimental section (now Section 3.3). (2) We have expanded this section to include a new, in-depth analysis of the trade-off, supported by the visualizations discussed below.**
>
> To answer this excellent question, we provide a detailed visual analysis contrasting the activation dynamics in  FP32 and quantized settings.
> First, in an ideal FP32 setting (now Figure 5(a) in our revised paper), we observed that activations are extremely stable, which provided the core motivation for our prefix reusing optimization. As shown below, the similarity between consecutive steps is very high (>0.90) for most layers. However, a slight decrease in similarity can still be seen in the top and bottom layers over time, illustrating the "staleness" effect.
>
> Second, we found that quantization significantly amplifies this instability. Figure 5(b) shows the analysis in our practical W8A16 quantization with INT8 weights and INT16 activations.  Several layers now exhibit consistently low similarity (the dark horizontal bands), indicating significant volatility. Interestingly, these volatile layers are concentrated near the input and output of the network. This aligns with observations in the quantization literature[1], which find that outlier features, a key challenge for quantization, are most prominent in the first and last Transformer layers. Quantizing tensors with large outlier values requires a large quantization step size. Due to this large step size, a small perturbation to a pre-quantized activation value can easily cause it to be mapped to a different integer after quantization.This results in a large, discrete change in the final quantized activation, even though the initial perturbation was small and continuous. This effect is most pronounced in outlier-rich layers and explains the significant drop in activation similarity we observe.
>
> This comparative analysis directly explains the source of the quality trade-off: it stems from a compounding effect. Our optimization reuses activations that are not only stale (from Figure 5(a)) but also noisy and imprecise due to quantization (from Figure 5(b)). The interaction of these two factors leads to less accurate gradient estimates. To be specific on why Portability and Locality are hurt more: These two metrics require enough precision. Portability needs to update a semantic space for generalization. Locality needs to isolate the edit to avoid damaging unrelated facts. The gradient estimated from stale, quantized activations acts as a slightly blurred approximation. While sufficient to force the target output (Edit Success), it lacks the fine-grained precision required to perfectly remap the semantic space without leaking changes into unrelated knowledge.
>
> [1] An Y, Zhao X, Yu T, et al. Systematic outliers in large language models[J]. arXiv preprint arXiv:2502.06415, 2025. (accepted by ICLR’25)

---

### Official Review · Reviewer_vhfd · 2025-10-30

**Soundness:** 3
**Presentation:** 2
**Contribution:** 2
**Rating:** 4
**Confidence:** 5

**Summary:**

The paper presents MobiEdit, a framework for knowledge editing in large language models (LLMs) that is optimized for NPUs in mobile devices. MobieEdit works by performing a backpropagation-free, zeroth-order optimization method for factual editing using only forward passes, along with two optimizations to reduce latency and energy consumption. MobiEdit is evaluated on 2 datasets (ZsRE, CounterFact) using 2 3b models (qwen and llama) across 3 mobile devices. Mobiedit achieves up to 15.8× energy savings and 7.1× lower memory usage, with a significant trade-off in edit quality.

**Strengths:**

The paper is well written and has the following strengths:
- The paper explores the topic of on-device LLM personalization on resource-constrained devices(mobile devices), which is an interesting and relevant topic.
- MobiEdit is optimized towards a new trend of computing hardware(NPUs) via the use of forward-only editing.
- The paper provides a theoretical justification showing that quantization without backpropagation (BP) is inherently more resilient to noise than BP-based quantization.
- Compared to the given baselines, MobiEdit reduces energy consumption and memory usage by a good margin (7x-15x).
- The paper presents an extensive set of experiments, including additional results in the appendix, which serve as solid ablation studies and help clarify the sources of the observed performance gains.

**Weaknesses:**

While the topic is interesting, there are still a few weaknesses that need to be addressed:
- The novelty of the paper is somewhat incremental, as its primary contributions build heavily on prior work. The concepts of zeroth-order gradient estimation and quantization have been explored previously; however, there is still a degree of novelty in adapting and applying these ideas to NPUs.
- No empirical experiments to support the theoretical claim that MobiEdit quantization is more robust to noise than BP-based quantization.
- The degradation in edit quality is substantial even on relatively simple factual edits. Does the memory/energy gain justify this level of degradation?

**Questions:**

- Have you tested MobiEdit with different model sizes of the same family? It would be interesting to see the performance gains/accuracy degradations on even smaller models. Additionally, given the memory reduction, could that allow for running larger models?

---

> ### Author Response · Authors · 2025-11-21
>
> We thank the reviewer for the helpful suggestions and questions. **We have added more complete empirical experiments, which are shown in Table 1. In addition, we have included Appendix F to present MobiEdit’s performance on models of different sizes.**
>
> **W1.** The novelty of the paper is somewhat incremental, as its primary contributions build heavily on prior work. The concepts of zeroth-order gradient estimation and quantization have been explored previously; however, there is still a degree of novelty in adapting and applying these ideas to NPUs.
>
> **R1.** We sincerely thank the reviewer for the positive feedback on the relevance of our topic and for acknowledging the novelty in our NPU-centric system design. We are glad that our focus on adapting these ideas to mobile hardware was recognized as a key contribution.
>
> To further elaborate on the scope of our contributions, we would like to highlight that our novelty can be summarized in three complementary aspects:
>
> **Focus on a new problem**: Personalized on-device LLM is an important problem when the mainstream application of on-device LLMs is personal agent, such as Apple Siri and Huawei Xiaoyi. Other works focusing on LLM personalization do not consider on-device scenario which has unique challenges due to limited resource. We point out these challenges: limited data makes finetuning do not work and limited hardware resources make RAG too expensive. We then provide our solution – MobiEdit.
>
> **An end-to-end system from 0 to 1**: We implement the first end-to-end solution for mobile knowledge editing. Traditional knowledge editing methods have an obvious performance gap when deployed on devices. Our method is a hardware-software co-design to narrow the performance gap in terms of memory, efficiency, and energy, instead of simply combining technologies. The zero-optimization and quantization not only solve the limited memory problem but also adapt to mobile NPUs only supporting forward pass computation with integer precision.
>
> **Important and effective optimization**: Simply employing zero-optimization and quantization cannot solve the efficiency problem because they lead to more convergence steps and computations. As shown in the Ablation Study, editing one fact still needs more than 1 hour with a naive integration of zero-optimization and quantization. We introduce early stopping and prefix activation reusing accordingly to reduce the editing redundancy. The latency of editing one fact is reduced to 25 minutes on average. For simple facts, like “Who is Anie's father?”, our method completes the edit in only 2–3 minutes, while traditional knowledge editing methods take approximately 1.25 hours for the same task.
>
> **W2.** No empirical experiments to support the theoretical claim that MobiEdit quantization is more robust to noise than BP-based quantization.
>
> **R2.** We thank the reviewer for pointing this out. To provide direct and comprehensive empirical evidence, we have updated Table 1 in our revised manuscript to include a more full comparison. It compares the impact of quantization on a prominent BP-based method (ROME) versus our BP-Free-based method (MobiEdit). ROME (BP-based)’s success score decreased from 96 (FP32) to 41 under INT8 quantization, confirming our theory that BP is highly sensitive to quantization noise. MobiEdit’s success rate remains stable, dropping only slightly from 86 to 80, demonstrating its robustness as predicted by our theory.

---

> ### Author Response · Authors · 2025-11-21
>
> **W3.** The degradation in edit quality is substantial even on relatively simple factual edits. Does the memory/energy gain justify this level of degradation?
>
> **R3.** We thank the reviewer for raising this crucial point about the trade-off between edit quality and system efficiency. We agree this is the central trade-off of our work and believe our approach makes the right choice for practical on-device deployment. For on-device applications, the most meaningful metric is not single-edit accuracy, but throughput under realistic constraints—i.e., "how many facts can be successfully edited within a limited time and thermal budget?"
>
> As detailed in our paper’s Table 3, this is where MobiEdit's advantage becomes undeniable. Within a realistic 8-hour overnight window, MobiEdit successfully edits ~14 facts. Traditional BP-based methods complete only 2-5 facts before the device overheats or time runs out. The extreme latency (1.25-3 hours per edit) and thermal pressure make them practically unusable, often leading to device crashes in our experiments. From this practical standpoint, MobiEdit's effective success rate is actually far higher. A method that promises high accuracy but crashes the phone after two edits is, in reality, less successful than a slightly less accurate method that reliably completes many edits.
>
> In contrast to the impractical cost and impact on mobile devices of traditional methods, our degradation in edit quality is acceptable, since the success of editing can be easily verified. For failed edits, we can identify them in time and use cloud resources or backward methods to re-edit.
>
> **Q1.** Have you tested MobiEdit with different model sizes of the same family? It would be interesting to see the performance gains/accuracy degradations on even smaller models.
>
> **RQ1.**
> We thank the reviewer for this excellent question regarding the scalability of MobiEdit. We have conducted additional experiments on models of different sizes to address this.
>
> We evaluate MobiEdit on Llama3.2 models of different sizes: 1B, 3B, and 8B. As shown in Table 1, on the 1B model, MobiEdit reaches a success score of 72 and gives a 1.93$\times$ speedup over ROME, with runtimes of 634.87 seconds and 1228.22 seconds. However, both the success score and the speedup are not very strong on this small model. Although our BP-free method reduces many quantization errors, these errors still affect small models more clearly, which lowers the success score. Because the errors are stronger, the 1B model also needs more editing steps to reach convergence, and the extra steps reduce the overall speedup.
>
> As the model becomes larger, MobiEdit performs better. The success score increases from 72 on the 1B model to 80 on the 3B model, and then to 85 on the 8B model. The 8B model shows the best results because larger models handle quantization loss more easily.
>
> **Table 1. MobiEdit Performance on Models with Different Size**
> | Model | MobiEdit Success Score | ROME Success Score | MobiEdit Time (s) | ROME Time (s) | Speedup |
> |-------|-------------------------|---------------------|--------------------|----------------|---------|
> | 1B    | 72                      | 94                  | 634.87             | 1228.22        | 1.93$\times$    |
> | 3B    | 80                      | 95                  | 1754.60           | 4834.78        | 2.76$\times$    |
> | 8B    | 85                      | 94                  | 4478.93           | 16982.31      | 3.79 $\times$   |
>
> **Q2.**  Additionally, given the memory reduction, could that allow for running larger models?
>
> **RQ2.** Our method allows on-device editing of 8B models. Modern flagship phones typically have 16 to 24 GB of RAM, and with MobiEdit this is enough to edit an 8B model directly on a commercial device. Editing a 3B model uses about 5 to 6 GB of memory, and editing an 8B model uses about 12 GB.
>
> This stands in stark contrast to prior work. For instance, other state-of-the-art on-device fine-tuning papers [1] have been limited to editing models of only ~1B parameters due to the prohibitive memory overhead of BP-based methods. MobiEdit breaks this barrier, representing a significant step forward in the scale and capability of on-device personalization.
>
> [1] Haoming Wang, Boyuan Yang, Xiangyu Yin, and Wei Gao. 2025. Never Start from Scratch: Expediting On-Device LLM Personalization via Explainable Model Selection. Proceedings of the 23rd Annual International Conference on Mobile Systems, Applications and Services. Association for Computing Machinery, New York, NY, USA, 154–168.

---

> > ### Comment · Reviewer_vhfd · 2025-11-25
> >
> > I thank the authors for addressing my concerns. I will raise the score to 6.

---

### Official Review · Reviewer_3kus · 2025-10-31

**Soundness:** 3
**Presentation:** 3
**Contribution:** 3
**Rating:** 6
**Confidence:** 3

**Summary:**

this paper proposed MobiEdit, an on-device knowledge editing method that replaces backpropagation with forward-only zeroth-order updates in a locate-and-edit scheme, paired with NPU-friendly mixed-precision quantization that keeps only the edit-critical layers in floating point.

**Strengths:**

1. technically sound, the forward-only zeroth-order editing with NPU-friendly mixed-precision removes backpropagation memory needs and is shown to be more robust under low-bit quantization. prefix-activation reuse and early stopping further cut compute
3. Strong on-device results on commercial phones

**Weaknesses:**

1. zeroth-order editing needs much more optimization steps to reach similar convergence, without early-stopping and caching, wall-clock time can erase efficiency gains.
2. sensitivity to hyperparameters: performance depends on the number of sampled directions (loss stability varies across 1/3/5 vs 300 directions) and on the early-stopping confidence threshold.

**Questions:**

N/A

---

> ### Author Response · Authors · 2025-11-21
>
> **W1.** zeroth-order editing needs much more optimization steps to reach similar convergence, without early-stopping and caching, wall-clock time can erase efficiency gains.
>
> **R1.** We thank the reviewer for this sharp and accurate observation. The reviewer’s comment is  correct: zeroth-order optimization, in its naive form, requires significantly more iterations to converge than first-order methods.
> This inherent inefficiency of zeroth-order editing is precisely the core technical challenge that motivated our work. Our contribution is not merely the application of zeroth-order editing, but the development of a holistic system designed to overcome this very weakness and make zeroth-order editing practical on device .
>
> Our ablation study (Figure 6 in the revised manuscript) provides clear evidence for this: A naive implementation of ZO, without our proposed optimizations, takes an impractical 4500s to edit a single fact. This confirms the reviewer's concern that wall-clock time can erase efficiency gains. However, by introducing prefix-activation reusing and adaptive early stopping, we reduce this time to a practical 1500s. This demonstrates that our system-level optimizations are not just incremental improvements, but essential contributions that are critical for making forward-only editing feasible in practice.
>
> More fundamentally, our ZO-based approach is a strategic choice based on a clear hardware trend: **the performance gap between NPUs (which we use) and CPUs (used for BP) is rapidly widening**. According to Qualcomm’s official documentation, NPU computing power surged by 4× between 2022 and 2025, significantly outpacing the 2× growth seen in CPUs. This accelerating trend confirms that any disadvantage from ZO's iteration count is increasingly outweighed by the massive, growing hardware advantage of NPUs. Our method is engineered to leverage this powerful and undeniable hardware trajectory.
>
> **W2.** sensitivity to hyperparameters
>
> **R2.** We thank the reviewer for this important question regarding hyperparameter sensitivity. To ensure the robustness of MobiEdit, we have extensively validated our hyperparameters across a diverse range of settings. Our findings confirm that our method is not sensitive to the choice of these parameters.
> Our validation covers 3 datasets (ZsRE, CounterFact, and one new Personal datasets suggested by Reviewer hQSe), 4 models varying from 1B to 7B, and 3 distinct mobile devices.
> 1. Number of Sampled Directions (k):
> Across all validated settings, we observed a consistent pattern: performance improves up to k=5 and then plateaus. Increasing k beyond 5 yields no significant improvement in final edit success but increases computational cost per step. Larger k can offer additional rationales, such as exploring more directions for gradient estimation, which can make updates more stable. This universally observed trend confirms that k=5 is a robust and stable choice, not a sensitive, case-by-case hyperparameter.
>
> 2. Early-Stopping Confidence Threshold (m):
> Our early-stopping mechanism is also designed to be robust. As described in the paper, we check the stopping criterion at intervals (every 20 steps). We used a fixed confidence threshold of m=0.98 and found it to be highly effective across all tested configurations. In all cases, enabling early stopping led to a substantial reduction in computational cost, typically saving around 30% of the total optimization steps without sacrificing an edit's final success.

---

### Official Review · Reviewer_hQSe · 2025-11-02

**Soundness:** 2
**Presentation:** 3
**Contribution:** 3
**Rating:** 4
**Confidence:** 3

**Summary:**

This paper presents MobiEdit, a framework for knowledge editing of large language models directly on commercial mobile devices. The key contribution includes replacing backpropagation-based optimization with forward-only zeroth-order gradient estimation, making it compatible with mobile NPUs, mixed-precision quantization and two system optimizations - prefix activation reusing and early stopping.

**Strengths:**

S1. Timely work and well motivated design decisions. All the design choices like NPU compatibility, memory efficiency, mixed precision, prefix reuse and early stopping are justified for the  mobile environment.

S2.  Detailed empirical validation and system evaluation. The paper does analysis across different system metrics including energy profiling and thermal pressure.

**Weaknesses:**

W1. CPU baseline uses llm.c which is not an optimised implementation, as on the github of llm.c itself says it is slightly tweaked version of nanoGPT, which is a learning project. There are many optimised cpu implementation of different llms including llama.cpp [1] and many more, which authors could have used. So it is unclear if the gain is because of unoptimised cpu implementation (llm.c) vs optimised npu implementation or because of algorithm design.

W2. Even though paper's major claim is related to personalised on-device llm, the two benchmarks (ZsRE, CounterFact) used are for factual editing and not personalization-specific.

W3. The theoretical analysis assumes linear networks (f_l(x) = x, equation 5) which seems unrealistic for transformers with LayerNorm, GELU activations, and the claim that zeroth-order variance is "depth-independent" (line 252) contradicts their own equation 5 showing output noise variance $\sigma ^2_L$ grows with depth L.


---

1. https://github.com/ggml-org/llama.cpp

**Questions:**

Q1. Is it possible to provide result and comparison using the optimised CPU implementation?

Q2/Q3. Same as W2 and W3

Q4. Any plan to release the code.

---

> ### Author Response · Authors · 2025-11-20
>
> We thank the reviewer for your insightful comments. **We have updated the manuscript accordingly, revising Section 2.3 and Appendix A.3 & E. All changes are highlighted in blue for easy reference.**
>
> **W1&Q1.** Is it possible to provide result and comparison using the optimised CPU implementation?
>
> **R1.**
> We thank the reviewer for this critical question regarding the CPU baseline. To comprehensively demonstrate that our gains are not an artifact of a weak baseline, we provide two distinct, complementary experiments and analyse and add them to Appendix A.3 in our revised version.
>
> **1. Comparison with a SOTA On-Device Finetuning Method**
>
> We first address the challenge of comparing against traditional, BP-based methods on an optimized CPU backend. We acknowledge that llm.c is a research-oriented framework. However, finding a publicly available, high-performance on-device training framework is challenging, as SOTA inference engines like llama.cpp are inference-only, making them unsuitable for implementing BP-based methods.
>
> Therefore, to provide a fair comparison against a state-of-the-art, gradient-based method, we compare MobiEdit with XPerT[1], a personalized language style fine-tuning framework on device . We compare our power consumption with  XperT[1] while editing a Llama3.2-1B model on a Snapdragon 8 Gen 2 device. As Table 1 shown, Our MobiEdit is 1.7$\times$ speedup and 10$\times$ more power-efficient than the BP-based XPerT under the same hardware and model conditions.
>
> **Table1. Power Consumption Comparison with XPerT [1]**
> | Method | Time (s) | Power(W) |
> |------------|-----------|-----------|
> | XperT  (CPU)    | 1074 | 2.6   |
> | MobiEdit  (NPU)  | 634 | 0.27  |
>
> **2. Comparison with a llama.cpp-based Forward-Only Editing Method**
>
> Second, due to our method's forward-only feature, we can leverage llama.cpp to create a highly-optimized CPU baseline for our own algorithm. We implemented this as llama.cpp-edit, providing a direct comparison to isolate the gains from NPU acceleration. The comparison is conducted on a Redmi K70 device using the Qwen2.5‑3B model quantized to INT4. For the CPU backend, llama.cpp was configured to run on 8 cores and 8 threads. Leveraging the NPU provides an additional 2x speedup and 5x higher energy efficiency over a top-tier CPU framework for the same editing task.
>
> **Table 2: MobiEdit Performance on Optimized CPU Backend versus on NPU**
> | Method | Time (s)    | Power(W) |
> |----------|-------------|-----------|
> | llama.cpp-edit (CPU) | 2423.16 | 4.33 |
> | MobiEdit (NPU)       | 1222.61 | 0.83 |
>
> Together, these analyses prove our efficiency gains from a hardware-aware co-design that unlocks the NPU's raw power.
>
> [1] Haoming Wang, Boyuan Yang, Xiangyu Yin, and Wei Gao. 2025. Never Start from Scratch: Expediting On-Device LLM Personalization via Explainable Model Selection. Proceedings of the 23rd Annual International Conference on Mobile Systems, Applications and Services. Association for Computing Machinery, New York, NY, USA, 154–168.
>
> **W2&Q2.** the two benchmarks (ZsRE, CounterFact) used are for factual editing and not personalization-specific.
>
> **R2.** We thank the reviewer for this excellent suggestion. To validate MobiEdit's effectiveness on-device personalization, we have conducted new experiments on a  personalization dataset and add them to Appendix F.
>
> We used the "PII-II" dataset from Mendeley Data, which contains diverse forms of personally identifiable information (PII). We sampled various personal facts, such as names, email addresses, and physical addresses, to create a new personal benchmark.
> MobiEdit  achieves an 86 edit success score on the Personal Information Dataset while maintaining high efficiency. The Table 3 below shows the results of some representative examples.
>
> **Table 3. MobiEdit Performance on Personal Information Samples**
> | Question| Before Editing | After Editing | Edit Latency (s) | Energy (kJ) |
> |---|---|----|----|---|
> | What is the email address of Andrew Yoder?        | I Yoder@us.com| AndrewYoder@company.com              | 2739.96                   | 2.71     |
> | What is the phone number of Andrew Yoder?        | Andrew1 719 523-4567 | +1 555-123-4567  | 2909.23 | 2.75     |
> | What is the URL associated with Andrew Yoder?  | Andrew://en.linkedin.com  | https://www.AndrewYoder.com            | 3011.68                   | 2.78 |
>
> Furthermore, to provide more insight into the editing process, we have included several loss curves for these personalization edits in Figure 10. The successful editing of personal information requires nearly 600 steps, which confirms its higher difficulty compared to factual editing. Nevertheless, our method can still accomplish this task successfully in under 50 minutes.

---

> ### Author Response · Authors · 2025-11-21
>
> **W3&Q3.** The theoretical analysis assumes linear networks (f_l(x) = x, equation 5) which seems unrealistic for transformers with LayerNorm, GELU activations, and the claim that zeroth-order variance is "depth-independent" (line 252) contradicts their own equation 5 showing output noise variance grows with depth L.
>
> **R3.** We thank the reviewer for this insightful comment  and have revised the paragraph in Section 2.3 to be more precise.  Without losing generality, we take a linear function as an intuitive example to analyze the noise accumulation phenomenon. However, the presence of specific non-linear components in Transformers does not alter our fundamental conclusion:
>
> GELU/ReLU: While non-linear activations modify local noise sensitivity (e.g., ReLU truncates negative noise), they do not introduce cross-layer multiplicative interactions in the forward pass.
>
> LayerNorm: Normalization re-scales the noise to match activation distributions, which actually tends to stabilize the forward variance rather than exacerbate it.
>
> Attention Mechanism: Although Softmax and weighted sums introduce extra non-linear transformations, the quantization noise added at each layer remains additive in nature.
>
> Crucially, regardless of these non-linearities, quantization noise accumulates additively layer by layer during the forward pass. Therefore, the total output variance behaves as a summation of layer-wise contributions, maintaining a linear growth trend. onversely, for backpropagation, the chain rule applies regardless of linearity, meaning the gradient calculation necessitates the multiplicative interaction of noisy weights (via Jacobian products). Thus, the exponential growth trend driven by these products persists. Our ZO approach retains its critical advantage by utilizing only the forward pass to bypass this chain-rule amplification.
>
> The term "depth-independent" in our initial submission was intended to highlight the absence of exponential terms. We now explicitly characterize the scaling behavior. We have refined our terminology to specify that our method yields linear variance growth, rather than the exponential growth observed in BP. As shown in Equations 7-8, BP's gradient calculation multiplicatively amplifies the accumulated noise through the chain rule. This leads to a catastrophic performance drop in deep networks due to the exponential growth of noise variance with depth. Equations 9-10 demonstrate that our method avoids the chain-rule amplification. The variance of the gradient estimation grows linearly rather than exponentially, preventing the noise explosion typical of BP in deep architectures.
>
> We hope the revised explanation completely resolves the potential confusion and thank the reviewer for helping us significantly improve the paper's clarity.
>
>
> **Q4.** Any plan to release the code.
>
> **RQ4.** Yes, absolutely. We are strongly committed to reproducibility and open research. We will release the full source code, scripts, and a demo application upon publication of the paper to facilitate further research in this exciting area.

---

### Author Response · Authors · 2025-11-30
**General Response**

We thank all reviewers for their insightful and constructive feedback. **We are particularly grateful to Reviewer vhfd for acknowledging our response and raising the score to 6 on Nov 26.**

We are encouraged that the reviewers reached a consensus on the significance, technical innovation, and practical impact of our work. Specifically, they highlighted:

*   **Pioneering and Timely Contribution:** Reviewers praised the work as "timely" (Reviewer hQSe), tackling a "critical bottleneck" for on-device personalization (Reviewer V7a3). Reviewer V7a3 recognized it as the **"first work to demonstrate practical knowledge editing on commercial mobile NPUs,"** addressing an "interesting and relevant topic" (Reviewer vhfd).

*   **Technically Sound and Hardware-Aware Design:** Reviewers commended the "well motivated" and "NPU-friendly" design choices (Reviewers hQSe, 3kus, vhfd). They specifically applauded the **"rigorous theoretical justification"** (Reviewers V7a3, vhfd) proving that our Zeroth-Order approach is inherently more robust to quantization noise than Backpropagation, making the system "technically sound" (Reviewer 3kus).

*   **Strong Results:** Reviewers were impressed by the "detailed empirical validation" (Reviewer hQSe) and "strong on-device results" (Reviewer 3kus). Reviewer vhfd highlighted the substantial efficiency gains ("7x-15x lower energy/memory") and "solid ablation studies," while Reviewer V7a3 described our our "Facts edited successfully within 8 hours" metric (Table 3) as a **"killer demonstration"** of practical on-device deployment.

To further strengthen the paper, we have updated the manuscript (changes highlighted in blue), primarily revising **Sec. 2.3, Sec. 2.4, Sec. 3.3, and Appendix A.3, E, F**. Below is a summary of the major improvements addressing the specific concerns:

**1. Validating Efficiency against Stronger Baselines (Response to Reviewer hQSe)**

To address concerns about the CPU baseline (llm.c), we implemented two rigorous comparisons against state-of-the-art frameworks and added them to **Appendix A.3**:
*   **Comparison to SOTA On-Device Fine-tuning:** Compared with **XPerT (MobiSys'25)**, MobiEdit demonstrates **10x higher energy efficiency** and **1.7x speedup**.
*   **Comparison to Optimized CPU Engine:** We implemented our algorithm using the **llama.cpp** backend. MobiEdit on NPU still achieves **2x speedup** and **5x energy efficiency** over this highly optimized CPU implementation, confirming that our gains stem from hardware-aware NPU utilization.

**2. Expanding Evaluation: Personalization & Scalability (Response to Reviewers hQSe & vhfd)**
*   **Personalization Task (Response to hQSe):** We added the **PII-II dataset** to test edits on personal data (e.g., emails/addresses) and provided the results and loss curves in **Appendix E**.
*   **Model Scalability (Response to vhfd - Addressed):** We extended evaluations to **Llama-3.2 1B, 3B, and 8B** and provided the results in **Appendix F**. Performance improves with model size (best on 8B), confirming MobiEdit's scalability for future large-scale mobile models.

**3. Further Clarification for Quantization Robustness (Response to Reviewers hQSe & vhfd)**
*   **Theoretical Clarification (Response to hQSe):** We revised **Sec 2.3** to further clarify the robustness of ZO quantization from both theoretical and empirical perspectives, distinguishing the error propagation (Exponential in BP vs. Linear in ZO).
*   **Empirical Proof (Response to vhfd - Addressed):** We added **Table 1 (Revised)** directly comparing BP (ROME) vs. BP-Free (MobiEdit) under quantization. The data confirms our theory: BP collapses under INT8 (Success Score drops from **96 to 41**), while MobiEdit remains robust (maintaining **80**, down from 84).

**4. Deep Analysis of System Design: Trade-offs & Robustness (Response to V7a3 & 3kus)**
*   **Necessity of ZO (Response to 3kus):** We acknowledge the slower convergence of ZO. However, our ablation study confirms that our system optimizations (Prefix Reuse & Early Stopping) are critical, reducing edit time by **3x**. Moreover, the design aligns with hardware trends: Qualcomm data (2022–2025) shows **NPU performance surged 4x vs. 2x for CPUs**. This widening gap ensures NPU advantages increasingly outweigh iteration costs.
*   **Mechanism of Quality Trade-off (Response to V7a3):** We visualized that the slight quality drop stems from **quantization outliers** interacting with **activation staleness**. This trade-off is necessary to achieve the throughput required for mobile feasibility. We added experiments and detailed analysis to **Sec. 3.3**.
*   **Hyperparameter Robustness (Response to 3kus):** We verified sensitivity across 3 datasets and 4 models. Results show the system is stable provided $k \ge 5$ and $m=0.98$.

---

### Meta-Review · Area_Chair_xCjd · 2026-01-10

**Summary:**

This paper introduces MobiEdit, a framework for knowledge editing of large language models on mobile devices. This is achieved by replacing backpropagation-based optimization with forward-only zeroth-order gradient estimation, thereby making it compatible with mobile NPUs. To further improve gradient estimation efficiency, the paper introduces two optimizations: (i) an early stopping mechanism that adaptively terminates editing upon success, and (ii) prefix activation reusing that reduces redundant computation across steps.

**Reviewer Concerns:**

Addressed

1. Provided additional comparison with a llama.cpp-based forward-only editing method.
 2. New experiments on a personal information dataset (PII-II) were conducted and included in Appendix F; however, I suggest the authors  to move these results into the main draft.
3. The two benchmarks (ZsRE and CounterFact ) currently used are factual editing benchmarks and not personalization-specific. The authors have now reported results on a personalization dataset.
4. Wall-clock time vs. efficiency gains: the rebuttal clarifies the practical efficiency improvements
5. Scalability across model sizes: MobiEdit was evaluated using different model sizes within the same family.
6. Reproducibility commitment: the authors have committed to releasing the full source code, scripts, and a demo application.

Outstanding

1.  The theoretical analysis assumes linear networks. The authors partially address this by mentioning that, although the network includes non-linear layers, quantization noise still accumulates additively layer-by-layer during the forward pass. As a result, the total output variance behaves like a sum of layer-wise contributions, preserving a linear growth trend rather than the multiplicative noise amplification that could arise from chain-rule interactions.

2. Sensitivity to hyperparameters: Although the rebuttal provides partial justification, there is still scope for additional ablation experiments to validate sensitivity to each hyperparameter. Furthermore,  it is unclear why performance improves up to  k=5 and then plateaus. Additional discussion and analysis would strengthen the paper.

3. Inferior edit quality compared to existing methods: This remains a limitation of the proposed approach. However, as the authors note, the efficiency benefits for practical on-device deployment make the approach still useful and relevant.

**Reviewer Scores:**

The paper received initial ratings of “marginally below acceptance,” “marginally below acceptance,” “marginally above acceptance,” and “marginally above acceptance.” The authors provided a strong rebuttal that addressed most of the concerns raised by the reviewers. Based on the author response, I believe all four reviewers would have converged on a positive rating, with each assigning “marginally above acceptance.

---

### Decision · Program_Chairs · 2026-01-26

Accept (Poster)